# Comparative primary paediatric nasal epithelial cell culture differentiation and RSV-induced cytopathogenesis following culture in two commercial media

**Lindsay Broadbent**[1], **Sheerien Manzoor**[1], **Maria C. Zarcone**[2], **Judit Barabas**[1], **Michael D. Shields**[1,3], **Sejal Saglani**[2], **Claire M. Lloyd**[2], **Andrew Bush**[4], **Adnan Custovic**[5], **Peter Ghazal**[6], **Mindy Gore**[2], **Ben Marsland**[7], **Graham Roberts**[8], **Jurgen Schwarze**[9], **Steve Turner**[10], **Ultan F. Power**[1] *

1 Wellcome-Wolfson Institute for Experimental Medicine, Queens University Belfast, Belfast, Northern Ireland, United Kingdom, 2 Inflammation, Repair and Development Section, National Heart and Lung Institute, Faculty of Medicine, Imperial College, London, England, United Kingdom, 3 Royal Belfast Hospital for Sick Children, Belfast Health & Social Care Trust, Belfast, Northern Ireland, United Kingdom, 4 Department of Paediatric Respiratory Medicine, Royal Brompton Hospital, London, England, United Kingdom, 5 Department of Paediatrics, Imperial College London, London, England, United Kingdom, 6 Division of Infection and Pathway Medicine, Deanery of Biomedical Sciences, University of Edinburgh Medical School, Edinburgh, Scotland, United Kingdom, 7 Department of Immunology and Pathology, Monash University, Melbourne, Australia, 8 Clinical and Experimental Sciences and Human Development and Health, Faculty of Medicine, University of Southampton, Southampton, England, United Kingdom, 9 Child Life and Health and MRC-Centre for Inflammation Research, Queen's Medical Research Institute, University of Edinburgh, Edinburgh, Scotland, United Kingdom, 10 Child Health, University of Aberdeen, Aberdeen, Scotland, United Kingdom

* u.power@qub.ac.uk

**Data Availability Statement:** Data available from a public repository run by Queen's University Belfast.

## Abstract

The culture of differentiated human airway epithelial cells allows the study of pathogen-host interactions and innate immune responses in a physiologically relevant *in vitro* model. As the use of primary cell culture has gained popularity the availability of the reagents needed to generate these cultures has increased. In this study we assessed two different media, Promocell and PneumaCult, during the differentiation and maintenance of well-differentiated primary nasal epithelial cell cultures (WD-PNECs). We compared and contrasted the consequences of these media on WD-PNEC morphological and physiological characteristics and their responses to respiratory syncytial virus (RSV) infection. We found that cultures generated using PneumaCult resulted in greater total numbers of smaller, tightly packed, pseudostratified cells. However, cultures from both media resulted in similar proportions of ciliated and goblet cells. There were no differences in RSV growth kinetics, although more ciliated cells were infected in the PneumaCult cultures. There was also significantly more IL-29/IFNλ1 secreted from PneumaCult compared to Promocell cultures following infection. In conclusion, the type of medium used for the differentiation of primary human airway epithelial cells may impact experimental results.

DOI: 10.17034/7c15ce87-a011-47e7-ba55-
50e0ccb550a9.

**Funding:** MDS, SS, CML, AB, AC, PG, BM, GR, JS,
ST and UFP were awarded grant number: 108818/
Z/15/B from The Wellcome Trust. https://wellcome.
ac.uk MDS and UFP were awarded grant number
Com/4044/09 from HSC Research & Development
(HSC R&D) Division. https://research.hscni.net;
https://wellcome.ac.uk/ The funders had no role in
study design, data collection and analysis, decision
to publish, or preparation of the manuscript.

**Competing interests:** The authors have declared
that no competing interests exist.

## Introduction

Eukaryotic cell culture is one of the fundamental techniques used by biomedical researchers.
Cell culture techniques are routinely used across most disciplines of life science research. Cell
culture has advanced dramatically in recent years with the development of differentiated primary epithelial cell cultures[1,2], organoids[3,4] and organ-on-chip systems[5,6]. One of the
key aspects of mammalian cell culture is the growth medium. The cell culture medium must
provide all of the key nutrients required for cell survival and division, an overview of which is
provided by Lodish et al[7]. The choice of cell culture medium is dependent on the cell type in
culture and the intended use of the cultures, as components of cell culture medium could affect
experimental outcomes. Primary cell culture and the development of more complex cellular
models requires highly specialised media to support the growth and differentiation of the cells.
This study focused on the culture of air-liquid interface differentiated primary airway epithelial
cells and their use in virus-host interaction research.

Advancements in airway epithelial primary cell culture, including the use of growth factors,
hormones and the use of an air-liquid interface[8,9], have led to important discoveries in virology and virus-host interactions[10–14]. The main advantage of using well-differentiated primary
airway epithelial cell cultures to study respiratory virus-host interactions is the similarity of the
cultures to the *in vivo* targets of infection. Well-differentiated primary airway epithelial cell
(WD-PAEC) cultures closely mimic the *in vivo* airways, demonstrating pseudostratified morphologies containing ciliated cells, mucus-producing goblet cells and tight junctions[2]. Indeed,
we previously demonstrated that WD-PAECs recreate several hallmarks of RSV infection *in
vivo*, including RSV infection of ciliated cells but not goblet cells, loss of ciliated cells, increased
goblet cell numbers, occasional syncytia, and the secretion of pro-inflammatory chemokines[15].

WD-PAEC cultures derived from patients with specific airway diseases often retain the features of that disease. Cultures derived from cystic fibrosis patients have been used to investigate
the potential for personalised treatment[16]. The differentiation of these cultures is essential for
measurement of CFTR function. The culture of primary airway epithelial cells has also enhanced
the diagnosis of primary ciliary dyskinesia, which is notoriously difficult to diagnose[17].

Initially, the choice of media for the culture of WD-PAECs was limited. However, as the use
of these culture systems increases in popularity the availability of specific reagents has also
increased. Our laboratory has cultured WD-PAECs for over ten years. Our protocols included
the use of Promocell Airway Epithelial Cell Growth Medium to differentiate and maintain the
cultures[2]. Using this method, we achieved over 90% success at differentiating primary airway
epithelial cell samples derived from paediatric nasal or bronchial brushes in Transwells. However,
for a period of ten months we experienced unexplained repeated failure in culture differentiation,
and our success rates decreased to ~50%. This led us to assess another primary cell medium,
PneumCult-ALI medium, for use in differentiating paediatric primary airway epithelial cells.

In this study, therefore, we evaluated the use of the two media in parallel for the culture and
differentiation of airway epithelial cells. We assessed the cultures for the total number of cells,
ciliated cells, goblet cells and epithelial integrity. We hypothesised that the choice of differentiation medium would affect the cytopathogenesis and antiviral immune responses of the
WD-PNEC cultures to RSV infection.

## Materials & methods

### Cell lines and viruses

The origin and characterization of the clinical isolate RSV BT2a were previously described [18].
RSV titres in biological samples were determined using HEp-2 cells, as previously described[19].

## WD-PNEC cultures

Primary nasal epithelial cells (n = 3 donors) were obtained from healthy paediatric patients with full parental consent. The nasal brushes were processed and the monolayer cell cultures were treated as previously described[2]. Cells were passaged twice in Promocell Airway Epithelial Cell Growth Medium (C-21160 Promocell) (supplements added as per the manufacturer's instructions with additional penicillin/streptomycin). When ~90% confluent the cells were seeded onto collagen coated Transwell supports (Corning) at $2x10^4$ or $5x10^4$ cells per Transwell. Cultures were submerged in modified Promocell Airway Epithelial Cell Growth Medium (see Table 1) supplemented with retinoic acid until fully confluent. After 4–6 days of submersion air-liquid interface (ALI) was initiated by removing the apical medium. This is required to trigger differentiation. Thereafter, half of the Transwell cultures were maintained in Promocell medium and half using PneumaCult-ALI medium supplemented with hydrocortisone and heparin. See Table 1 for constituents of the media, where known. Stemcell Technologies, the producer of PneumaCult, did not disclose the ingredients of the supplements provided with the medium. Medium was replaced with 500 μL of fresh medium in the basolateral compartment every 2 days. Complete differentiation took at least 21 days. Cultures were only used when hallmarks of excellent differentiation were evident, including extensive apical coverage with beating cilia and obvious mucus production. Trans-epithelial electrical resistance (TEER) was measured using an EVOM2 and ENDOHM 6 mm chamber (World Precision Instruments).

## Infection

WD-PNECs were infected apically for 2 h at 37°C with $1.4x10^5$ $TCID_{50}$ RSV BT2a in 50 μL of DMEM (low glucose, no additives). Cultures were then rinsed 4 times with 250 μL DMEM (low glucose, no additives). The fourth wash was retained as the 2 hpi time point. At 24 hpi and every 24 h thereafter until 96 hpi apical washes were undertaken and harvested by adding 250 μL DMEM apically, pipetted up and down gently and aspirated without damaging the cultures, added to cryovials and snap frozen in liquid nitrogen. RSV titres in biological samples were determined by a tissue culture infectious dose 50 (TCID50) assay, as previously described [19].

## Immunofluorescence

WD-PNECs were fixed with 200 μL apically and 500 μL basolaterally of 4% PFA (v/v in PBS) for 1 h then permeabilised with 0.1% Triton X-100 (v/v in PBS) for 1 h. Cells were blocked

**Table 1. Known constituents of promocell and PnemaCult differentiation media.**

| Promocell Airway Epithelial Cell Growth Medium | | | PneumaCult-ALI Medium | | |
|---|---|---|---|---|---|
| Promocell kit supplements | BPE | 52 μg/mL | PneumaCult- ALI x10 supplement | Unknown | Unknown |
| | hEGF | 10 ng/mL | | | |
| | Insulin | 5 μg/mL | | | |
| | Hydrocortisone | 0.5 μg/mL | PneumaCult-ALI maintenance supplement x100 | Unknown | Unknown |
| | Epinephrine | 0.5 μg/mL | | | |
| | Transferrin | 10 μg/mL | | | |
| User-optimised supplements | BSA | 1.5 μg/mL | User-optimised supplements | Hydrocortisone | $1x10^{-6}$ M |
| | Retinoic acid | 15 ng/mL | | Heparin | 4 μg/mL |
| | Penicillin | 100 U/mL | | Penicillin | 100 U/mL |
| | Streptomycin | 100 μg/mL | | Streptomycin | 100 μg/mL |

with 0.4% BSA (v/v in PBS) for 30 min. Immunofluorescent staining was performed for Muc5Ac (1:100 dilution, mouse monoclonal; Abcam) (goblet cell marker), β-tubulin (1:200 dilution, rabbit polyclonal Cy3 conjugatedl; Abcam) (ciliated cell marker) and RSV F protein (1:500 dilution, 488 conjugated; Millipore). Cultures were mounted using DAPI mounting medium (Vectashield, Vector Labs) and imaged using a Nikon Eclipse 90i or a Leica SP5 confocal microscope. For ZO-1 images cultures were fixed in 4% PFA for 20 min at room temperature, followed by permeabilization (Permeabilization Buffer set, Ebioscience) and blocking with 2% BSA solution (Sigma). Cells were stained with anti-ZO-1 mouse mAb (Thermo-Fischer, Alexa Fluor 488). Images were acquired on an inverted laser scanning confocal microscope (SP5, Leica Microsystems).

### IFNλ1/IL-29 ELISA

The concentration of IFNλ1/IL-29 was measured in basolateral medium from RSV BT2a- or mock-infected cultures at 96 hpi by ELISA (Thermo Fisher Scientific; BMS2049). The manufacturer's instructions were followed.

### Microscopy and image analysis

For enumeration of cell types, a minimum of 5 fields were captured per condition/well per patient by UV microscopy (Nikon Eclipse 90i). Differential interference contrast (DIC) microscopy was used to capture bright field images of differentiated cultures. Image analysis was carried out using ImageJ software (http://rsbweb.nih.gov/ij/). ImageJ was also used to calculate the diameter of cells. The diameter of >40 cells across 5 fields of view per patient were measured.

### Statistical analysis

GraphPad Prism $^®$ was used to create graphical representations of the data and for statistical analyse. To assess statistical significance results were compared using t tests, except for viral growth kinetics, which were compared by calculating the areas under the curves.

## Results

To determine the effect of Promocell or PneumaCult medium on cell proliferation during differentiation, cells were seeded at two different densities on Transwell supports. Trans-epithelial electrical resistance (TEER), a measure of epithelial integrity, was measured in cultures seeded with $5x10^4$ cells. There was a trend towards increased TEERs in PneumaCult cultures but this did not reach significance (Fig 1A). Expression of ZO-1, a marker of tight junctions, was clearly evident in cultures differentiated in both media (Fig 1C). Cultures were trypsinised to determine the total cell count (Fig 1B). The seeding density, either $2x10^4$ or $5x10^4$ cells per Transwell, did not affect the final number of cells in the cultures. PneumaCult medium resulted in ~3-fold higher cell counts following differentiation than Promocell medium.

Cells differentiated in Promocell appeared larger than those in PneumaCult under light and fluorescent microscopy. This was confirmed by imaging the cultures using DIC microscopy and measuring the cell diameters (Fig 2A and 2B). Cells differentiated and maintained in Promocell medium were significantly larger than cells in cultures from the same donors but differentiated using PneumaCult medium. Confocal microscopy revealed that the cells within the PneumaCult cultures appeared more tightly packed (Fig 2C). Orthogonal sections suggest a greater degree of stratification of the cultures differentiated in PneumaCult medium compared to Promocell medium.

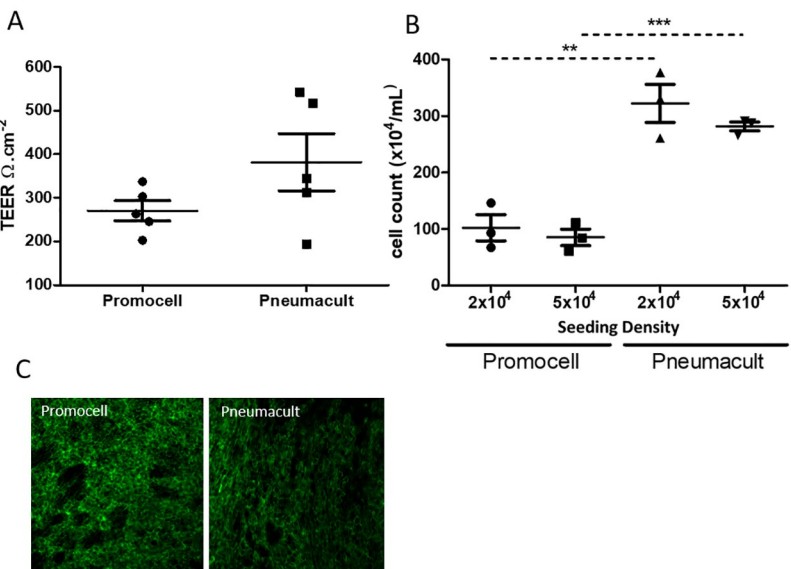

**Fig 1. Primary paediatric nasal epithelial cells were passaged twice then seeded on collagen coated Transwell supports at a seeding density of 2x104 or 5x104 per Transwell.** Cells were maintained in Promocell medium while submerged. Following ALI initiation half of the cultures from each donor were maintained using Promocell and half using Pneumacult. Cultures were differentiated for 21 days. TEER values were measured in the cultures seeded at 5x104 (n = 5 donors) (A). Cultures were trypsinised and a cell count was performed (n = 3 donors) (B). Cultures were fixed in 4% PFA and stained for ZO-1 (n = 3), representative images (C). Statistical significance was determined using unpaired t-tests. ** = $p < 0.01$, *** = $p < 0.001$.

The number of total, ciliated and goblet cells in fixed cultures differentiated in Promocell or PneumaCult medium were enumerated from *en face* images following fluorescent microscopy of cultures stained for DAPI (nuclei), β-tubulin (cilia) and Muc5Ac (goblet cells), respectively. Representative images of cultures from both media are presented in Fig 3A. Consistent with the data presented in Fig 1B above, the PneumaCult-maintained cultures demonstrated higher numbers of total cells, as well as ciliated and goblet cells (Fig 3B). In all cases, there was a trend towards increased cell numbers in the PneumaCult cultures, although they did not reach significance. When the proportion of ciliated and goblet cells was calculated, however, there was no difference between Promocell- or PneumaCult-maintained cultures (Promocell: 75.3% ciliated and 5.1% goblet cells; PneumaCult: 75.1% ciliated and 4.2% goblet cells). This was consistent for both seeding densities (data not shown for seeding density $2x10^4$) (Fig 3C).

A central theme of research in our laboratory is to study RSV interactions with paediatric airway epithelium. To explore whether the medium used affected RSV growth kinetics or cytopathogenesis, cultures were infected with the low passaged clinical isolate RSV BT2a. The same amount of virus ($1.4x10^5$ TCID$_{50}$) was inoculated onto all cultures. At the specified times post infection, apical washes were titrated on HEp-2 cells to determine virus growth kinetics (Fig 4). There was no significant difference in viral growth kinetics between the two initial seeding densities of the cultures or the medium used to differentiate and maintain the cultures. As RSV infects ciliated epithelium and, because of higher ciliated cell numbers, we expected the PneumaCult cultures to reach higher peak viral titres released from them. However, all culture conditions resulted in similar peak viral titres and growth kinetics.

The secretion of IFNλ1/IL-29, a type-III interferon known to be the main interferon secreted following RSV infection of airway epithelium [20,21], was quantified in the basolateral medium at 96 hpi (Fig 6). There was significantly more IFNλ1/IL-29 secreted from the PneumaCult compared to the Promocell cultures. This may be due to the larger number of cells

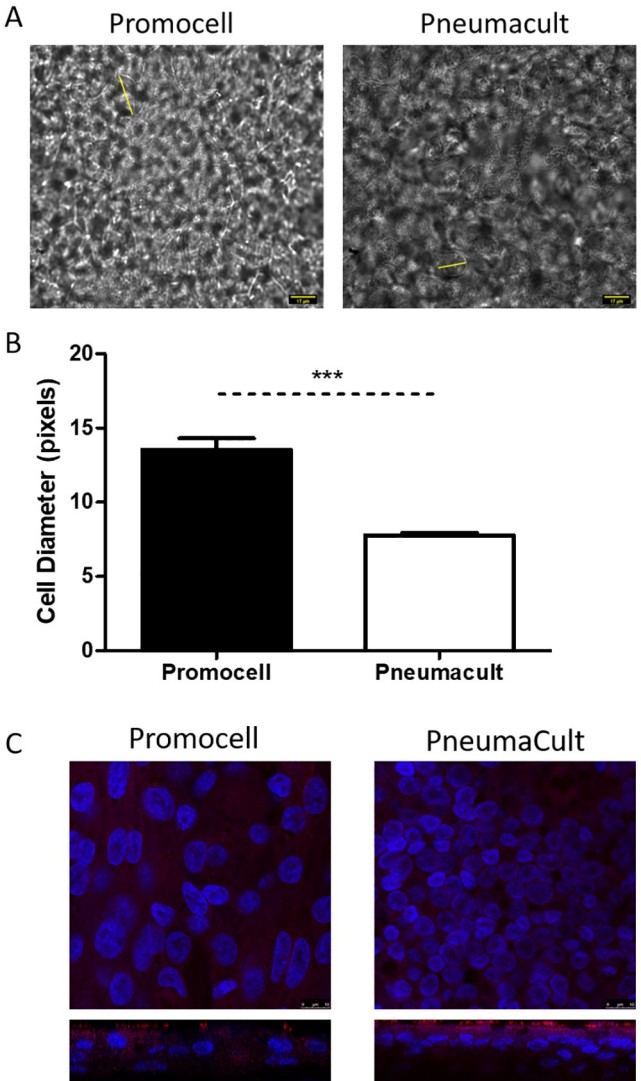

**Fig 2. Primary paediatric nasal epithelial cells (n = 3 donors) seeded at 5x10⁴ per Transwell were differentiated and maintained using either Promocell or Pneumacult medium.** Cultures were fixed using 4% paraformaldehyde (PFA) on day 25 post air-liquid interface (ALI) initiation. Images were captured by DIC microscopy at x60 magnification and imageJ was used to determine the diameter of cells (yellow lines) (A). Graphical representation of the average cell diameter in pixels(B). Statistical significance was determined using unpaired t-tests. *** = p<0.001. Cultures were stained for beta-tubulin (red) and DAPI (blue). Z-stacks were obtained using a confocal microscope at x100 magnification (Leica SP5) (C).

present in the PneumaCult cultures. However, there are approximately 3x the number of cells in the PneumaCult compared to the Promocell cultures, yet the IFNλ1/IL-29 was >6x that secreted from Promocell cultures. The PneumaCult cultures may respond more robustly to infection but further work, investigating different cytokines/chemokines, would be needed to provide further insights into these differences in innate immune responses to RSV infection.

## Discussion

In this study we confirmed our hypothesis that the choice of medium affects the resultant cultures. PneumaCult medium resulted in cultures with ~3x more cells than those differentiated

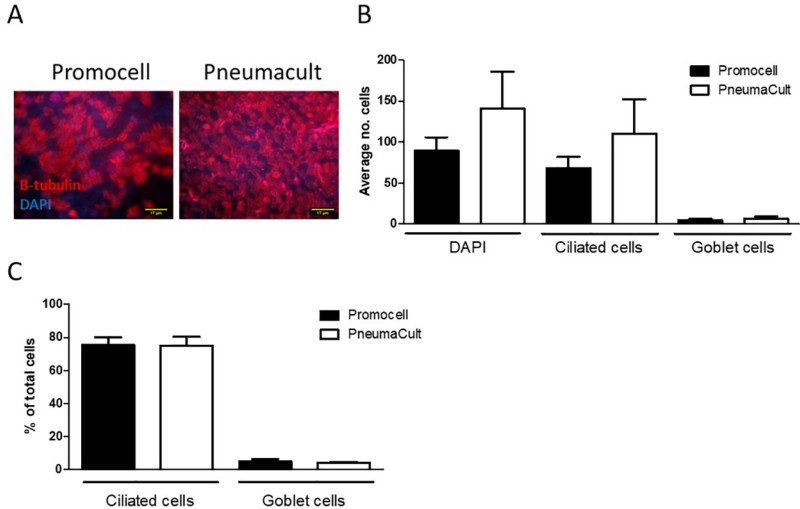

**Fig 3. WD-PNEC cultures (n = 3 donors) with an initial seeding density of $5\times10^4$ per Transwell were differentiated in Promocell or Pneumacult medium.** After 21 days cultures were fixed in 4% paraformaldehyde and stained for β-tubulin, a ciliated cell marker; Muc5ac, a goblet cell marker and counterstained for DAPI. Representative images of β-tubulin staining (A). The average number of total, ciliated and goblet cells from 5 fields of view per donor was calculated (B). The percentage of ciliated cells and goblet cells in the culture was calculated (C). Images were acquired using a Nikon Eclipse 90i at x60 magnification.

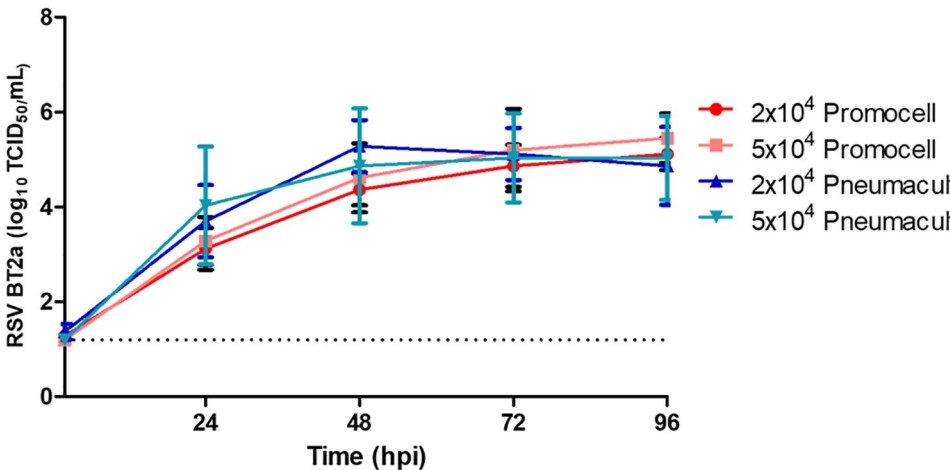

**Fig 4. Primary paediatric nasal epithelial cells (n = 3 donors) seeded on collagen coated Transwell supports at a seeding density of $2\times10^4$ or $5\times10^4$ per Transwell.** Were maintained in either Promocell or Pneumacult medium for 21 days. Cultures were infected with RSV BT2a $1.4\times10^5$ $TCID_{50}$. Apical washes were harvested at 2 and 24 hpi and every 24 h thereafter and titrated on HEp-2 cells to determine virus growth kinetics. RSV-infected cultures were fixed at 96 hpi and the total number of cells and the number of ciliated, goblet and RSV-infected cells were enumerated in *en face* IF images (Fig 5). Following infection, the mean number of cells was significantly different between Promocell- and PneumaCult-differentiated cultures, with a 46% and 37% reduction in mean cell numbers, respectively. All cultures demonstrated a similar loss in ciliated cell numbers following RSV infection, 20% and 23% reduction for Promocell and PneumaCult, respectively. There were significantly more RSV-infected cells in the PneumaCult cultures, consistent with higher numbers of ciliated cells in these cultures compared to the Promocell cultures. However, despite the higher numbers of ciliated cells, as previously mentioned (Fig 3B), and the greater number of RSV-infected cells (Fig 5B), the viral growth kinetics were not significantly different. The percentage of ciliated, goblet and RSV-infected cells in the cultures did not differ significantly as a function of the culture medium used.

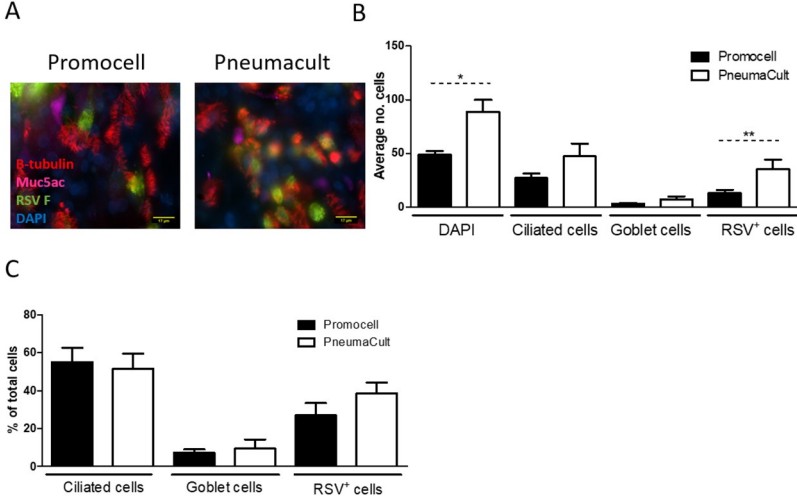

**Fig 5. WD-PNEC cultures (n = 3 donors) were differentiated in Promocell or Pneumacult medium.** After 21 days cultures were infected with RSV BT2a or mock infected. At 96 hpi the cultures were fixed in 4% paraformaldehyde and stained for β-tubulin, a ciliated cell marker; Muc5ac, a goblet cell marker, RSV F and counterstained for DAPI. Representative images of β-tubulin staining (A). The average number of total, ciliated, goblet and RSV infected cells from 5 fields of view per donor was calculated (B). The percentage of ciliated, goblet and RSV infected cells in the culture was calculated (C). Images were acquired using a Nikon Eclipse 90i at x60 magnification. Statistical significance was determined by t-test.

using Promocell medium. Interestingly, despite the differences in total cells counts, the proportions of ciliated cells and goblet cells were similar for both culture conditions. Indeed, the proportion of ciliated cells was consistent with the proportion reported in normal healthy human airway epithelium (50–70%) [22,23]. The limited evidence available suggests that

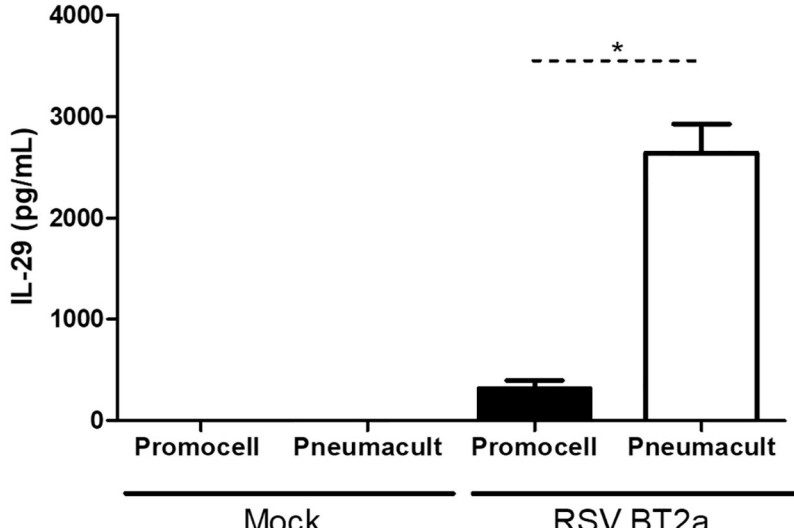

**Fig 6. WD-PNEC cultures (n = 3 donors) were differentiated in Promocell or Pneumacult medium.** After 21 days cultures were infected with RSV BT2a or mock infected. Basolateral medium was harvested and snap frozen every 24 hpi. The concentration of IFNλ1/IL-29 in the basolateral medium at 96 hpi was quantified by ELISA. Statistical significance was determined by t-test. * = p<0.05.

goblet cells represent up to 25% of cells in adult human airway epithelium[24]. The proportions of goblet cells found in our current cultures were considerably lower than this. However, we are unaware of the lower proportion of goblet cells found in normal human paediatric airway epithelium and, therefore, cannot conclude whether or not our cultures are abnormal with respect to goblet cell content. The percent goblet cells that we report here were also lower compared to our previous WD-PNEC cultures derived from newborn and 1-year-old infants [25]. Reasons for these discrepancies remain to be elucidated, although they may be due to nasal epithelial cell donor- or culture-specific factors.

Ciliated cells, mucus and the airway surface liquid (ASL) are key components of the mucociliary escalator, the primary defence mechanism against inhaled pathogens and foreign particulate material. RSV infection disrupts this by having a detrimental effect on the number of ciliated cells[26]. RSV-induced cilia loss was replicated in this WD-PNEC model under both culture conditions, reinforcing the evidence that the WD-PAEC model reliably recreates at least some RSV cytopathogenesis.

Another noticeable impact of the different media was the different cell sizes. There is very little published data available on the size of human nasal epithelial cells *in vivo*. Due to experimental differences, it is difficult to compare cultured nasal epithelial cells. However, previous work from our group demonstrated differentiated nasal epithelial cells of ~12 μm in diameter (data not shown). In the present study cells of ~14 μm and ~25 μm diameters from PneumaCult and Promocell differentiation media, respectively, were evident.

As we are unaware of the precise constituents of the proprietary PneumaCult medium, it is not possible to determine which components, if any, might be responsible for the differential cell count and size between the cultures. We are aware, however, that Promocell medium contains bovine pituitary extract (BPE) (52 μg/mL), while PneumaCult medium apparently does not. BPE contains components which are needed for differentiation of epithelial cells, including growth factors and hormones[27]. As it is derived from animal tissues, the components of BPE can vary between batches. This may explain in part the variable success rates we previously experienced in differentiating WD-PNEC cultures from nasal brushes. However, this remains to be confirmed. BPE-free media, such as PneumaCult, bypass the need for this component by supplementing with a cocktail of hormones and growth factors, allowing for greater reproducibility of the composition of different batches of media.

Although the proportion of ciliated cells is the same in both cultures there was an increase in actual numbers of ciliated cells in the PneumaCult cultures. Ciliated cells are the primary target for RSV infection[15,28]. As such, following RSV infection there were more RSV infected cells in the PneumaCult compared with the Promocell cultures. Both media resulted in cultures that were successfully infected with RSV with very similar viral growth kinetics to that previously reported [2,15]. The difference in cell numbers and, indeed, the difference in the number of RSV+ cells in the cultures did not have a significant impact on apically-released virus titres. This indicates that the number of cells within a culture is not a defining factor in viral growth kinetics. The factors affecting viral growth kinetics are not fully understood. Interestingly, much higher concentrations of IFNλ1/IL-29 were secreted from RSV-infected PneumaCult cultures, which may be due to the higher cell density within the culture. We previously demonstrated that IFNλ1/IL-29 was responsible for attenuating RSV growth kinetics in WD-PBECs[21]. The higher IFNλ1/IL-29 concentrations secreted from the RSV-infected PneumaCult compared to Promocell cultures, therefore, might explain in part the similarities in RSV growth kinetics, despite the higher number of RSV+ cells in the former cultures.

In conclusion, both media tested under these conditions result in WD-PAEC cultures that possess several hallmarks of airway epithelium *in vivo* and resulted in comparable experimental outcomes in several of the parameters assessed. However, our data also indicate that the

choice of medium used to differentiate and maintain primary airway epithelial cell cultures may impact the experimental outcomes and care should be taken in choosing medium for the intended work. However, one should be cognisant of the low donor numbers used in this study. It should also be noted that we did not use either media 'off-the-shelf' and extensive optimisation is often needed to achieve the best culturing conditions. As cell culturing techniques advance and become more sophisticated there will undoubtedly be an increase in reagents created specifically for this purpose. As such, it will be imperative that independent comparisons between different reagents, such as media, are undertaken to ensure reliability of the data generated.

## Author Contributions

**Conceptualization:** Lindsay Broadbent, Maria C. Zarcone, Claire M. Lloyd, Ultan F. Power.

**Data curation:** Lindsay Broadbent, Maria C. Zarcone.

**Formal analysis:** Lindsay Broadbent.

**Funding acquisition:** Michael D. Shields, Sejal Saglani, Claire M. Lloyd, Andrew Bush, Adnan Custovic, Peter Ghazal, Ben Marsland, Graham Roberts, Jurgen Schwarze, Steve Turner, Ultan F. Power.

**Investigation:** Lindsay Broadbent, Sheerien Manzoor, Judit Barabas.

**Methodology:** Lindsay Broadbent, Sheerien Manzoor, Sejal Saglani, Claire M. Lloyd, Ultan F. Power.

**Project administration:** Mindy Gore.

**Supervision:** Ultan F. Power.

**Visualization:** Michael D. Shields.

**Writing – original draft:** Lindsay Broadbent.

**Writing – review & editing:** Lindsay Broadbent, Sheerien Manzoor, Maria C. Zarcone, Michael D. Shields, Sejal Saglani, Claire M. Lloyd, Andrew Bush, Adnan Custovic, Peter Ghazal, Mindy Gore, Ben Marsland, Graham Roberts, Jurgen Schwarze, Steve Turner, Ultan F. Power.

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
