## [Decision Letter · Decision Letter 0]

29 Jan 2020

PONE-D-20-00512

Comparative primary paediatric nasal epithelial cell culture differentiation and RSV-induced cytopathogenesis following culture in two commercial media.

PLOS ONE

Dear Dr Broadbent,

Thank you for submitting your manuscript to PLOS ONE. After careful consideration, we feel that it has merit but does not fully meet PLOS ONE’s publication criteria as it currently stands. Therefore, we invite you to submit a revised version of the manuscript that addresses the points raised during the review process.

The study examines a small number subjects (n=3) and the experiment does not appear to be replicated as noted by a reviewer. The issues noted by the reviewers must be addressed, particularly the consistency of results among plates and among different days (replicates). 

We would appreciate receiving your revised manuscript by Mar 14 2020 11:59PM. To enhance the reproducibility of your results, we recommend that if applicable you deposit your laboratory protocols in protocols.io, where a protocol can be assigned its own identifier (DOI) such that it can be cited independently in the future. For instructions see: http://journals.plos.org/plosone/s/submission-guidelines#loc-laboratory-protocols

We look forward to receiving your revised manuscript.

Kind regards,

Ralph A. Tripp

Academic Editor

PLOS ONE

Journal Requirements:

Additional Editor Comments:

The study examines a small number subjects (n=3) and the experiment does not appear to be replicated as noted by a reviewer. The issues noted by the reviewers must be addressed, particularly the consistency of results among plates and among different days (replicates).

Reviewers' comments:

Reviewer's Responses to Questions

**Comments to the Author**

1. Is the manuscript technically sound, and do the data support the conclusions?

Reviewer #1: Yes

Reviewer #2: Partly

2. Has the statistical analysis been performed appropriately and rigorously? 

Reviewer #1: Yes

Reviewer #2: No

3. Have the authors made all data underlying the findings in their manuscript fully available?

Reviewer #1: Yes

Reviewer #2: Yes

4. Is the manuscript presented in an intelligible fashion and written in standard English?

Reviewer #1: Yes

Reviewer #2: Yes

5. Review Comments to the Author

Reviewer #1: This study assessed two different culture media, Promocell and PneumaCult during the differentiation and maintenance phases of isolated human primary nasal epithelial cell cultures. Their impacts on the morphological and physiological characteristics and responses to exposure to respiratory syncytial virus exposure were measured. It was shown that PneumaCult promoted greater numbers of small, tightly packed cells that appeared in a pseudostratified arrangement. Both media treatments gave similar proportions of ciliated and goblet cells, with no detectably significant differences in RSV growth kinetics. Interestingly more ciliated cells were infected in the PneumaCult cultures. The authors concluded that the media used in the culture of primary human airway epithelial cells is an important consideration.

The methods and procedures were appropriate and the study essentially well executed. The authors have published methodological papers on this subject previously (see ref 2), and provided excellent morphologic/physiologic data to support their findings. The figures provided in the current study, though informative, were not of the quality presented in the earlier work. This is particularly true of the conclusion drawn that a greater degree of (pseudo?) stratification of the cultures differentiated was achieved in the PneumaCult medium compared to Promocell (data not shown). This may be a particularly important point given the subject matter and approach in this study.

There were concerns about the small number subject sampled (n=3), which was duly noted (line302) and unknown proprietary constituents in Pneumacult media, which is not trivial (lines 273-283). But it is important for those in the field to provide these types of comparisons in order to ensure confidence and reproducibility of similar approaches.

Reviewer #2: This is a relatively simple study that has relevance to the field of airway in vitro assays. The strengths are that multiple donors we used and several measured endpoints were conducted. The important finding is that both culture systems generated many non-significant effects as pointed out including RSV titers. The weakness is that the manuscript emphasizes the few differences. Of these differences proliferation rate was a main difference based on seeding density but this did not affect functional endpoints. And one cannot conclude it is proliferation rate because plating efficiency was not monitored. In other word some plates may had fewer cells initially attach to the matrix. But this was not measured. Lastly although there were a number of donors, the experiment does not appear to be replicated. Thus the consistency of results among plates and among different days (replicates) was either not measured or not performed.

The above points should be addressed with more than one replicate and Z factor scores calculated to actually understand whether the differences seen were due to differences between media type or just random variation.

6. PLOS authors have the option to publish the peer review history of their article (what does this mean?). If published, this will include your full peer review and any attached files.

Reviewer #1: No

Reviewer #2: Yes: Steven Stice

---

## [Author Response · Author response to Decision Letter 0]

4 Mar 2020

Response to editor’s and reviewer’s comments:

Editor: The study examines a small number subjects (n=3) and the experiment does not appear to be replicated as noted by a reviewer. The issues noted by the reviewers must be addressed, particularly the consistency of results among plates and among different days (replicates). 

Response: We recognise that the main point of concern for the editor and reviewers is the number of donors used and the replicates. We understand that n=3 is limited and ideally, we would increase this. However, it should be noted that these primary nasal epithelial cells are from neonatal donors within 10 days of life and, as such, are incredibly precious samples with limited number of cells. The use of these cultures, and indeed every Transwell, had to be meticulously reasoned. As the reviewer pointed out we have stated that this is a limitation of the study in the discussion. 

We would also like to draw the editor’s attention to the reproducibility of the results within the different culture conditions, despite the expected donor-to-donor variation. In addition, the data from Transwells seeded at the two different seeding densities (2x104 and 5x104) were remarkably similar for cell counts and virus growth kinetics at the time of exploitation for these experiments (Figure 1B and Figure 4 of the manuscript) (Figure 1 in 'response to reviewers document'). Included below is the uninfected Transwell immunofluorescence data to highlight the similarities between the two seeding densities. 

We have also included the ELISA data for IL-29 secretion following mock or RSV BT2a infection for Transwells derived from both seeding densities to show the remarkable similarity between these results (Figure 2 in 'response to reviewers document'). We are very confident, therefore, that these data are reproducible within our experimental conditions. We chose to present only the data derived from the seeding density of 5x104 for some of the figures for simplicity and consistency of experimental conditions. We have mentioned this is the revised manuscript (Line 189). However, if the Editor and/or Reviewers require us to include the extra data from Transwell cultures seeded at 2x105 cells, we would be happy to modify to manuscript accordingly.

Reviewer 1 states: ‘This is particularly true of the conclusion drawn that a greater degree of (pseudo?) stratification of the cultures differentiated was achieved in the PneumaCult medium compared to Promocell (data not shown).’

Response: We have modified the language (now Line 171) to avoid over emphasis on the differences in orthogonal sections to read: ‘Orthogonal sections suggest a greater degree of stratification of the cultures differentiated in PneumaCult medium compared to Promocell medium. Understanding the apparent greater stratification would require much more extensive work and would constitute a new project/manuscript. 

Reviewer 1 states: ‘unknown proprietary constituents in Pneumacult media, which is not trivial (lines 273-283).’

Response: We thank the reviewer for emphasising this point and have stated this is the manuscript. 

Reviewer 2 states: ‘The important finding is that both culture systems generated many non-significant effects as pointed out including RSV titers. The weakness is that the manuscript emphasizes the few differences.’

Response: We have modified the language in the discussion to indicate that there are several similarities between cultures grown using the different media. The paragraph now reads (line 300): ‘In conclusion, both media tested under these conditions result in WD-PAEC cultures that possess several hallmarks of airway epithelium in vivo and resulted in comparable experimental outcomes in several of the parameters assessed. However, our data also indicate that the choice of medium used to differentiate and maintain primary airway epithelial cell cultures may impact the experimental outcomes and care should be taken in choosing medium for the intended work’

Reviewer 2 states: ‘one cannot conclude it is proliferation rate because plating efficiency was not monitored. In other word some plates may had fewer cells initially attach to the matrix. But this was not measured.’ 

Response: We apologise that this may have been unclear in the original submission, but the cell cultures from all donors were seeded at the same density in Promocell medium. Only at the time of air-liquid interface (ALI) initiation were half of the cultures (per patient) grown in Pneumacult medium. Therefore, the number of cells that adhere to the Transwells should be virtually identical as the seeding conditions are exactly the same. Additionally, the two different seeding densities do not affect the ultimate number of cells in the differentiated cultures. This indicates that Pneumacult continues to promote cell growth and proliferation post ALI, which we think is a very important point. 

Reviewer 2 states: ‘Thus the consistency of results among plates and among different days (replicates) was either not measured or not performed.’

Response: Experimentation on each donor was performed independently and were not run in parallel. There is remarkable consistency in these experiments, even with the expected donor-to-donor variation as shown in figures 1B and 4 and in the additional data included above. Please also see the response the editor’s main point. We are confident, therefore, that the differences arising in our data, although minor, are due to the different media and are not random variations. 

Reviewer 2 states: ‘The above points should be addressed with more than one replicate and Z factor scores calculated to actually understand whether the differences seen were due to differences between media type or just random variation.

Although not ideal for small data sets such as this, we have calculated the z factor scores (in 'response to reviewers' document). Although not strong due to small sample size, the z factor scores for total cell number and ELISA reinforce our statistically significant findings comparing Promocell and Pneumacult media. The negative z factor scores for other figures is indicative of non-significant and overlapping results between the different test groups.

---

## [Editor Report · Decision Letter 1]

6 Mar 2020

Comparative primary paediatric nasal epithelial cell culture differentiation and RSV-induced cytopathogenesis following culture in two commercial media.

PONE-D-20-00512R1

Dear Dr. Broadbent,

We are pleased to inform you that your manuscript has been judged scientifically suitable for publication and will be formally accepted for publication once it complies with all outstanding technical requirements.

With kind regards,

Ralph A. Tripp

Academic Editor

PLOS ONE

Additional Editor Comments (optional):

The authors have satisfactorily addressed the reviewers.
---

## [Editor Report · Acceptance letter]

10 Mar 2020

PONE-D-20-00512R1 

Comparative primary paediatric nasal epithelial cell culture differentiation and RSV-induced cytopathogenesis following culture in two commercial media. 

Dear Dr. Broadbent:

I am pleased to inform you that your manuscript has been deemed suitable for publication in PLOS ONE. Congratulations! Your manuscript is now with our production department. 

With kind regards,

on behalf of

Dr. Ralph A. Tripp 

Academic Editor

PLOS ONE